# Pharmacy-Based Opportunistic Atrial Fibrillation Screening at a Community Level: A Real-Life Study

**DOI:** 10.3390/healthcare10010090

**Published:** 2022-01-04

**Authors:** Stephane Olindo, Pauline Renou, François Martial, Nathalie Heyvang, Lea Milan, Sylvain Ledure, François Rouanet

**Affiliations:** 1Stroke Unit Department, University Hospital of Bordeaux, Place Amelie Raba-Leon, CEDEX, 33076 Bordeaux, France; pauline.renou@chu-bordeaux.fr (P.R.); nathalie.heyvang@chu-bordeaux.fr (N.H.); lea.milan@chu-bordeaux.fr (L.M.); sylvain.ledure@chu-bordeaux.fr (S.L.); francois.rouanet@chu-bordeaux.fr (F.R.); 2Union Régionale des Professionnels de Santé Pharmaciens Nouvelle-Aquitaine, 33000 Bordeaux, France; francois.martial@resopharma.fr

**Keywords:** atrial fibrillation, screening, pharmacy, community, elderly

## Abstract

Purpose: Opportunistic pharmacy-based screening of atrial fibrillation (AF) appears effective, but the proportion of detected citizens is unknown. The aim of our real-life study was to determine rates of screening in a community population according to age group and gender. Methods: We conducted four community campaigns of pharmacy-based single-time point AF screening that involved individuals ≥65 years. We used a single-lead and hand-held device MyDiagnostick (6229 EV Maastricht, The Netherlands) that generates a 60-s ECG trace. All pharmacies of the communities (*n* = 54) were involved. Rates of screening were assessed on the base of the French National Institute for Statistics and Economic Studies data and were expressed as percentage and 95% Confidence interval (CI). Results: We screened 4208 individuals (Mean age, 74.2 ± 6.6 years; females, 60.2%). The screening rate in citizens aged ≥65 years was 17.2% (16.6–17.7), and higher in females than in males (17.9% [17.3–18.6] versus 16.0 [15.3–16.8], *p* < 0.001). The 70–74 age group showed the highest rate (25.7% [24.4–27]) compared to other groups. After 74 years, screening rates decreased steadily with age and dropped to 4.8% [3.8–6.1] in very elderly (≥90). Among the 188 (4.47%) positive screening, 117 (2.78%) showed an AF that was unknown in 53 (1.26%). Increasing age (OR: 1.05 [1.00–1.09], *p* = 0.04), male sex (OR: 4.30 [2.33–7.92], *p* < 0.0001) and high CHA2DS2-Vasc (OR: 1.59 [1.21–2.09], *p* = 0.0008) were independent predictors of unknown AF. Conclusion: Single-lead AF detection performed in community pharmacies result in screening one in six elderly citizens. Although male sex and elderly predicted unknown AF diagnosis, they were less involved in such designed campaigns.

## 1. Introduction

Atrial fibrillation (AF) is the most common sustained arrythmia and is characterized by a growing prevalence that increases steadily with age [1]. The disease is closely associated with stroke [2], and diagnosis of AF is of the utmost importance since anticoagulation treatment is safe and efficient in preventing ischemic events [3]. Unfortunately, AF is often clinically silent and is frequently underdiagnosed [4]. The European Society of Cardiology (ESC) recommend an opportunistic screening by pulse taking or ECG in patients ≥65 years of age [3]. In recent years, development of mobile health technologies provides opportunities to screen large group of population. Several studies assessed AF detection tools such as watches [5], smartphones [6] or hand-held devices [7]. These trials have been performed in primary care center, geriatric ward or pharmacies [8]. Efficiency in new AF detection deeply differ (0.7% to 9.5%) between opportunistic single-time screening [9] and multi-time screening in selected individuals [10]. Screening tools that record single-lead ECG strip of ≥30 s may be particularly useful when they allow ECG trace reviewing by a physician and eventually AF diagnosis [3].

In this report, we assessed our real-life experience of large campaigns of opportunistic AF screening in community pharmacies that focused on unselected customers aged ≥65 years.

## 2. Methods

### 2.1. Aim of the Study

The purpose of our study was to assess a real-life procedure of AF screening involving community pharmacists, general practitioners (GPs) and cardiologists.

### 2.2. Design of the Screening Campaigns and Procedure

Screening campaigns resulted from an initiative of our Stroke Unit and were logistically supported by the URPS Pharmaciens (Union Régionale des Professionnelles de Santé Pharmaciens) and by town councils of the communities. The ARS (Agence Régionale de Santé) Nouvelle Aquitaine funded the project.

Protect-AVC was a single-time-point AF screening implemented in all pharmacies of 4 communities located in the area of Bordeaux. The 4 screening campaigns were sequentially performed from October 2018 to March 2020.

Fifty-four pharmacies agreed to participate. In the 2 weeks preceding campaigns initiation, the population of the communities was informed through local newspapers, town council newsletters, flyers and radio spots that an AF screening was available in their pharmacies. GPs and cardiologists of the campaign area were systematically informed about the procedure of the AF screening.

A face-to-face presentation of the screening procedure was performed with the pharmacists in each participating pharmacy. Briefly, the procedure was as follows:

If they wished to, individuals aged ≥65 years could perform an AF screening in one of the community pharmacies involved in the campaigns. When they had available time, pharmacists could also invite their pharmacy customers to participate to the campaign. With the pharmacist assistance, individuals were asked to fill a short questionnaire that focused on demographic characteristics and on the presence of vascular risk factors such as high blood pressure, diabetes, heart failure, peripheral vascular disease, history of TIA or stroke, known AF, anticoagulation treatment and presence of an implantable pacemaker. The questionnaire was deidentified and only the first letters of participant name and surname were recorded. The name of their GP was also requested and individuals were asked to tick one box if they were against the transmission of the detection result to their GP or against the use of the collected data.

The AF screening was performed with MyDiagnostick device, a hand-held single-lead ECG (Applied Biomedical Systems BV, 6229 EV Maastricht, The Netherlands). The record takes 1 min and analysis conclusion is immediately displayed by turning either green for normal cardiac rhythm or red for AF detection.

In case of positive AF screening, the captured 60 s ECG trace was downloaded to the computer and printed by the pharmacist. The participating individual was aware of the need to visit his GP with an envelope containing an information letter, the printed ECG and a return-form questionnaire in a pre-stamped envelope. The GP was also systematically kept informed by post that his patient has participated to the Protect-AVC campaign and has been diagnosed with a probable AF. In France, the GP is the coordinator of the patient pathway and if required he refers his patient to the cardiologist. Thus, the decision to continue cardiac investigations such as a 12-lead ECG, or a 24 h ECG holter was left to the GP and cardiologist’s discretion.

Through the return questionnaire, the GP informed the study team of his conclusion as follows: AF previously unknown, AF previously known, and AF not confirmed.

Data of all filled participant questionnaires, results of the MyDiagnostick screening and of the return questionnaire were collected.

### 2.3. Analysis of the Positive Mydiagnostick ECG Traces

For all positive screenings, MyDiagnostick ECG trace patterns were analyzed by the two investigators (F.R. and S.O.). Assessment resulted in a classification as follows: AF, sinus rhythm, sinus arrythmia, extrasystoles, motion artefact or undetermined. In case of disagreement, a definitive classification was adjudicated by consensus. Arrythmia and absence of *p* wave determined AF.

For the study assessment, an unknown AF was defined as: (1) Typical AF on ECG strip and (2) absence of previous AF confirmed by the GP or lack of knowledge of AF by the participant without any anticoagulation treatment at the time of screening.

## 3. Statistical Analysis

Our screening population was stratified by age (65–69, 70–74, 75–79, 80–84, 85–89 and ≥90 years). The numerator for calculation of screening rate was the number of participants detected in an age group and the denominator was based on the estimation of age structure of the population provided by French National Institute for Statistics and Economic Studies (INSEE). Data were collected on 10 February 2020 through the www.insee.fr link. Age structure data was available for Pessac, Arcachon and Saint-Medard en Jalles (SMEJ) communities. Proportion of screening was expressed as a percentage and the exact 95% binomial confidence intervals (CI) was used.

The χ^2^ test and Student t test were used to examine differences in nominal and continuous values. *p*-value of <0.05 was considered statistically significant.

A multivariable logistic regression analysis was performed to identify factors predictive of previously unknown AF. Factors of potential significance in univariate analysis (defined as *p* < 0.1) were introduced into the multivariable model in a backward stepwise manner. Independent factors were expressed as Odd Ratio and 95% CI.

## 4. Individuals Consents

Individuals were informed of their anonymized participation in this research, and the possibility to withdraw was offered by ticking a box in the short questionnaire that was systematically filled before MyDiagnostick screening.

## 5. Data Availability

Anonymized database used for the current analysis will be available upon written request to the corresponding author from any qualified investigator.

## 6. Results

The four campaigns have screened 4208 individuals aged ≥65 years old. Periods of screening of the four campaigns, numbers and demographic characteristics of detected individuals are summarized in Table 1. The mean age of screened participants was 74.2 ± 6.6 years old and women were prevalent (60.1%).

Size and age structure of the population were available in three communities, Pessac, Arcachon and SMEJ, and Figure 1 shows the age pyramid for ≥65 years old people. In the pooled population (≥65 years) of the three communities (*n* = 21,521), 3694 subjects have participated to the campaign giving a total screening rate of 17.2% (16.6–17.7). The rate was significantly higher in female than in male (17.9% [17.3–18.6] versus 16.0 [15.3–16.8], *p* < 0.001). Rates of screening in groups of population aged between 65–69, 70–74, 75–79, 80–84, 85–89 and ≥90 years old and according to gender are described in Table 2 and illustrated in Figure 2 and Figure 3. The 70–74 group showed the significant highest rate (25.7% [24.4–27]) compared to other groups. Over 74 years old, rates of screening decreased steadily, and the oldest aged group (≥90 years old) showed the lowest rate of detection (4.8% [3.8–6.1]). The screening rates were significantly higher in females than in males in the 65–69- and 70–74-years old groups, 18.8% (17.5–20.1) versus 14.3% (13.0–15.6) and 30.1% (28.2–32.0) versus 20.6% (18.9–22.4), *p* < 0.001, respectively. The proportions of population that have been detected in each campaign are described in Appendix A.

The Pessac campaign was performed over the longest period of time (10.5 months) compared to the three other campaigns which screened individuals between 3.1 and 3.8 months. The rate of detection in the Pessac population was similar to those found in Arcachon or SMEJ and 83% of the total screening was performed during the first 4.5 months.

A total of 188 (4.47%) individuals were screened positive. The analysis of the single-lead ECG generated by MyDiagnostick device revealed a typical AF pattern in 117 cases (2.78%), extrasystoles in six cases (0.14%), motion artefacts in 56 cases (1.33%) and undetermined in nine cases (0.21%). Among the 117 AF individuals, 53 (1.26%) had an unknown AF and 64 suffered from a known AF (1.52%) (Figure 4). A total of 70 (1.7%) participants reported wearing an implanted pacemaker. In this subgroup, 13 (18.6%) were screened positive including 10 with a previously known AF and three positive screening related to artifacts. No previously unknown AF was detected in subjects with implanted pacemakers. In the whole studied population, the rates of unknown AF detection tend to increase with age group, but the differences were only significant for the comparison between 65–69 and older age groups (Appendix A).

Compared to the negative screening group, unknown AF individuals were older (77.7 ± 6.2 vs. 73.9 ± 6.5, *p* < 0.0001) and predominantly males (61.5% vs. 39.3%, *p* < 0.0001). They showed a higher prevalence of diabetes (18.9% vs. 9.5%, *p* = 0.03) and history of heart failure (15.9% vs. 6.3%, *p* = 0.02) and had a higher CHA2-DS2-Vasc score (3.17 ± 1.03 vs. 2.68 ± 1.05, *p* = 0.001). Multivariable analysis found that increasing age (OR: 1.05, 95% CI [1.00–1.09], *p* = 0.04), male sex (OR: 4.30, 95% CI [2.33–7.92], *p* < 0.0001) and high CHA2DS2-Vasc (OR: 1.59, 95% CI [1.21–2.09], *p* = 0.0008) were independent predictors of unknown AF (Table 3).

In summary, the main findings of the study are: (1) Our procedure resulted in screening 17.2% (16.6–17.7) of individuals ≥65 years of a community population. (2) The 70–74 years age group showed the highest AF screening rate (25.7% [24.4–27]), then rates steadily declined with increasing age until 4.8% (2.2–7.4) in the very elderly population (≥90 years old). (3) The screening rate was significantly lower in males than in females in the 65–69- and 70–74-years age groups, 14.3% (12.5–16.0) versus 18.8% (17.3–20.3) and 20.6% (18.6–22.6) versus 30.1% (28.4–31.9), respectively. 4) Rates of previously unknown AF diagnosed during the campaigns was 1.26% (0.93–1.62).

## 7. Discussion

We report real-life data of large campaigns of opportunistic AF screening setting in community pharmacies. The screening campaigns focused on unselected customers aged ≥65 years old and detection was performed by MyDiagnostick, a hand-held single-lead ECG device.

We found a previously unknown AF in 1.26% of screened individuals. This rate is in line with results disclosed in a recent meta-analysis [8] of studies using single-lead ECG devices or Holter monitoring. The mean AF screening rate was assessed at 1.7%. In single-time point detection studies, the rate was around 1% ranging from 0.5% [9] to 5.3% [11] in a selected group of individuals with a large proportion of post-stroke condition. In accordance with previous studies [12,13], individuals with new AF were more likely to be older and male, to have medical history of diabetes and of heart failure, and to score a higher CHA2DS2-Vasc. Old age, male-sex and higher CHA2DS2-Vasc score constituted independent factors of unknown AF detection.

Several single-lead AF screening trials have been performed in community pharmacies (Table 4) [6,7,14,15,16,17]. However, no data is available regarding the proportion of people detected in a population community included in an AF screening pharmacy program. We found that AF detection campaigns setting in pharmacies allowed to screen one in six (17.2%) individuals of a community aged ≥65 years. This proportion reached one in four (25.7%) subjects in the 70–74 years age group. The rate decreased steadily with age, and the age group ≥90 years showed the lowest rate (4.8%). It may reflect that oldest population poorly visit the community pharmacies due to mobility impairment. Additionally, increasing age is associated with nursing home living due to medical condition. Although very elderly individuals are most at risk of AF, they are poorly involved in community pharmacy screening program. Alternative strategies in very elderly should be based on at home screening design.

Whereas male sex was an independent predictor of unknown AF detection, men were less involved than women in the pharmacy screening program (16.0% [15.3–16.8] versus 17.9% [17.3–18.6]). It was particularly significant for the youngest population with a difference that reached 10% for the 70–74 years age group (20.6% [18.9–22.4] versus 30.1% [28.2–32.0]). Reasons for the gender difference are not clear. It is admitted that compared to women, men have a lower health literacy that is defined as skills to promote and maintain good health through access to, understanding and use of specific information [18]. Campaigns of prevention such as influenza vaccination also show that males are less involved than females [19]. According to social habits, men may visit pharmacies less often to collect treatment or to buy drugstore items.

In our study, rates of screened population did not differ between the three community campaigns. Interestingly, the longest campaign in the Pessac community did not screen a higher proportion of population than the shortest campaigns in the Arcachon and SMEJ communities. Therefore, we assume that repeated short-time campaigns of detection may be preferred to continuous screening.

AF screening campaigns in community pharmacy may constitute an effective primary prevention strategy mostly in the 65–79 years old population. Indeed, distribution of pharmacies throughout the national territory allows large AF screening in general population particularly in non-urban areas where shortage of medical doctor is increasingly significant. Additionally, compared with pulse palpation or no detection, opportunistic AF screening implemented in pharmacies has been considered to be cost-effective in UK [16]. On the other hand, screening disease induces anxiety in participants and the psychological harm results in being labelled with an unexpected disease diagnosis [20]. Pharmacists who participate to the screening campaigns may need to strengthen their knowledge on pathophysiology and on communication particularly in disclosing information according to the detection result.

## 8. Limitations of the Study

The present work has several limitations. Among the 117 participants who showed an AF on the single-lead ECG trace, a definitive conclusion by the GP was obtained for only 68 (58.1%). However, unknown AF made sense since it was retained in participants who were not treated with anticoagulation and not aware of AF. Assessment of screened population proportion did not take into account the exact address location of the participants and proportion may have been overestimated. Conversely, individuals who live at the municipality border may used to visit pharmacies of another community and then underestimate the screening rate. Screening rates reported in our study could not be applied in other countries with different healthcare organization. The strength of our study is based on collection of prospective data from large real-life campaigns with involvement of all pharmacies of the participated communities.

## 9. Conclusions

Real-life screening campaigns setting in community pharmacies allow to identify previously unknow AF in 1.26%. The population aged between 65 and 79 years old is the preferred target. The procedure is associated with a screening gap for elderly individuals and males. Alternative design of detection such as at home screening should be developed for elderly individuals and strategy of information focusing on male population may be proposed.

## Figures and Tables

**Figure 1 healthcare-10-00090-f001:**
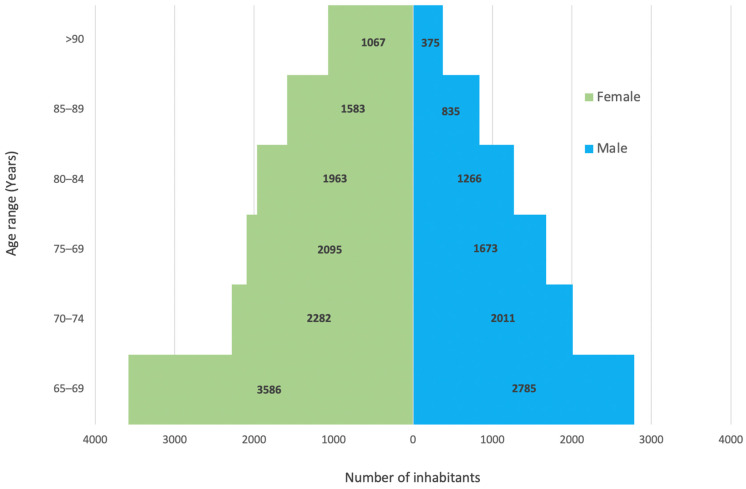
Age pyramid of the pooled population ≥65 years old of the 3 screened communities.

**Figure 2 healthcare-10-00090-f002:**
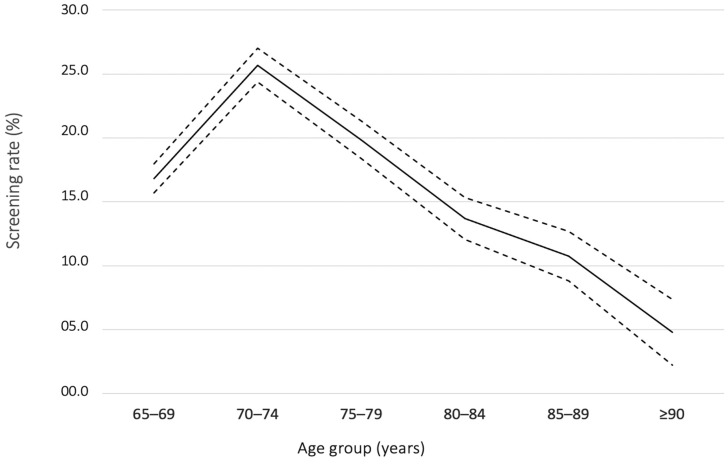
Atrial Fibrillation screening rates according to age group for all participants. The continuous line represents the value of the screening rate in percentage and dotted lines show the upper and lower limits of the 95% confidence interval.

**Figure 3 healthcare-10-00090-f003:**
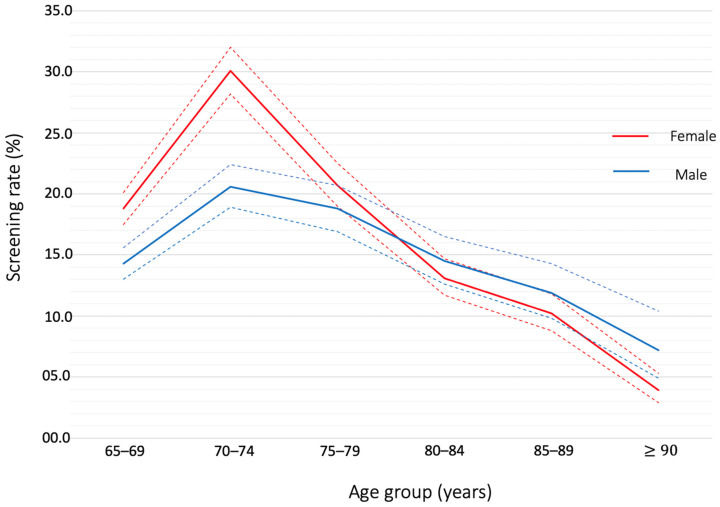
Atrial Fibrillation screening rates according to age group and gender. The continuous line represents the value of the screening rate in percentage and dotted lines show the upper and lower limits of the 95% confidence interval.

**Figure 4 healthcare-10-00090-f004:**
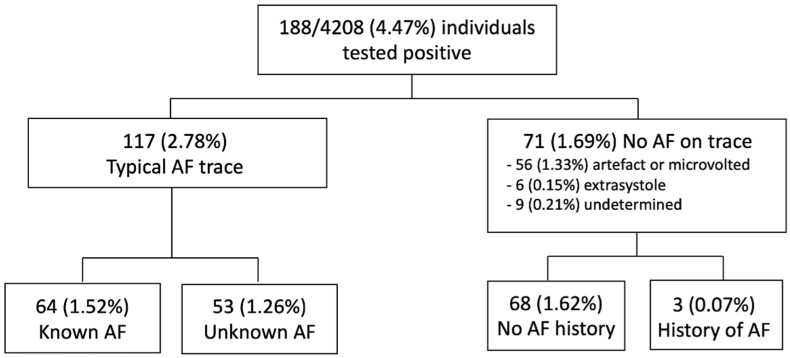
Flow chart of participants in the study.

**Table 1 healthcare-10-00090-t001:** Characteristics of the 4 campaigns of atrial fibrillation screening.

	The 4 Communities	Pessac	North Medoc	Arcachon	Saint-Medard En Jalles
Screening period (date)	18 October 2018 to 15 April 2020	18 October 2018 to 31 August 2019	13 September 2019 to 15 December 2019	10 September 2019 to 05 January 2020	04 January 2020 to 15 April 2020
Screening period (months)	-	10.4	3.1	3.8	3.4
Number of pharmacies involved (*n*)	54	18	14	11	11
Individuals ≥ 65 y.o. screened (*n*)	4208	1570	514	1124	1000
Mean number of screenings per month	-	148.1	165.8	288.2	294.1
Mean number of screenings per pharmacy	77.9	87.2	36.7	102.2	90.9
Population ≥ 65 y.o. in the community (*n*)	-	10,143	-	5503	5875
Age, y.o., mean ± Standard deviation (Range)	74.2 ± 6.6 (65–100)	74 ± 6.6 (65–100)	74 ± 6.2 (65–100)	74.3 ± 6.8 (65–100)	73.6 ± 6.3 (65–95)
Female (%)	60.2	62.1	52.5	60.5	60.8
Positive Screening, *n* (%)	188 (4.47)	65 (4.14)	31 (6.03)	60 (5.33)	32 (3.20)
Positive Screening in relation with Atrial Fibrillation *n* (%)	117 (2.78)	38 (2.42)	19 (3.69)	38 (3.38)	22 (2.20)

**Table 2 healthcare-10-00090-t002:** Rates of screening according to age group and gender in the pooled population of 3 communities.

Age Group (Years)	All	Female	Male	*p*
65–69	16.8 (15.9–17.8)	18.8 (17.5–20.1)	14.3 (13.0–15.6)	<0.001
70–74	25.7 (24.4–27.0)	30.1 (28.2–32.0)	20.6 (18.9–22.4)	<0.001
75–79	19.9 (18.6–21.2)	20.7 (19.0–22.5)	18.8 (16.9–20.7)	0.120
80–84	13.7 (12.5–14.9)	13.1 (11.7–14.7)	14.5 (12.6–16.5)	0.280
85–89	10.8 (9.6–12.1)	10.2 (8.8–11.8)	11.9 (9.8–14.3)	0.200
≥90	4.8 (3.8–6.1)	3.9 (2.9–5.3)	7.2 (4.9–10.4)	0.016
≥65	17.2 (16.6–17.7)	17.9 (17.3–18.6)	16.0 (15.3–16.8)	<0.001

Rates of screening are expressed as percentage and 95% Confidence Intervals. Statistical Comparison was performed between female and male groups.

**Table 3 healthcare-10-00090-t003:** Characteristics of positive screening with unknow AF and negative screening participants. Uni- and multi-variable analysis.

	Unknown AF Participants N = 53	Negative Screening Participants N = 4020	Univariable Analysis		Multivariable Analysis	
			Odd Ratio (95% CI)	*p*	Odd Ratio (95% CI)	*p*
Age (years), mean ± SD	77.7 ± 6.2	73.9 ± 6.5	1.08 (1.04–1.12)	<0.0001	1.05 (1.00–1.09)	0.0400
Male	35 (66.0)	1584 (39.4)	2.99 (1.69–5.30)	0.0002	4.30 (2.33–7.92)	<0.0001
Arterial Hypertension	27 (50.9)	1757 (43.7)	1.34 (0.78–2.30)	0.29		
Diabetes	10 (18.9)	380 (9.5)	2.23 (1.11–4.47)	0.02		
History of Stroke	3 (6.1)	186 (4.6)	1.26 (0.39–4.09)	0.69		
History of Heart failure	7 (15.9)	240 (6.3)	2.39 (1.07–5.36)	0.03		
Peripheral arterial disease	5 (10.2)	177 (4.6)	2.26 (0.89–5.75)	0.08		
CHA2DS2Vasc	3.17 ± 1.03	2.68 ± 1.05	1.49 (1.18–1.88)	0.0008	1.59 (1.21–2.09)	0.0008

Nominal values are expressed as number (percentage). AF: Atrial Fibrillation.

**Table 4 healthcare-10-00090-t004:** Summary of trials investigating AF detection using single-lead ECG devices and setting in pharmacies.

Study	Country	N	Individuals	Device	Mean/Median Age (Years)	Females (%)	New AF (*n*)	Rate of New AF (%)
Lowres et al. (2014)	Australia	1000	Unselected	Alive Cor	76 ± 7	56	15	1.5
Twigg et al. (2016)	United Kingdom	594	Unselected pharmacy customer	Alive Cor	68.3 ± 8.9	-	5	0.8
Alves da Costa et al. (2020)	10 countries	1741	Unselected pharmacy customer	Alive Cor	69.6 ± 13	67.3	5	0.57
Savickas et al. (2020)	United Kingdom	604	>65 years and attending an influenza vaccination	Alive Cor	73 (69–78)	57.3	4	0.7
Zaprutko et al. (2020)	Poland	525	Unselected pharmacy customer > 65 years	Alive Cor	73.7 ± 6.5	68.2	7	1.33
Zink et al. (2021)	Germany	7107	Unselected pharmacy customer > 65 years	MyDiagnostick	74 ± 5.9	58	256	3.6

AF: Atrial Fibrillation.

## Data Availability

Data that support the study results could be obtained upon request from the following email: stephane.olindo@chu-bordeaux.fr.

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
