# Peer review of "Pharmacy-Based Opportunistic Atrial Fibrillation Screening at a Community Level: A Real-Life Study"

_healthcare, 2022, doi:10.3390/healthcare10010090_

Round 1

Reviewer 1 Report

Thank you very much for inviting me to the manuscript review. This is an interesting population study on the prevalence of atrial fibrillation in the population of 4 regions of France. Atrial fibrillation is a common arrhythmia.As it may occur asymptomatically, it is dangerous due to the risk of ischemic strokes in patients without anticoagulant treatment. Therefore, knowledge of the presence of asymptomatic atrial fibrillation is of great importance for patients and health care.The use of the screening method in a pharmacy with the use of a network of pharmacies, where a large number of patients end up, is ingenious. I have some comments to improve the quality of the manuscript

1. Complete, if possible, size and age structure of the North Medoc population. This information is important for a complete picture of the study group.

2. If possible, create an age pyramid of the studied population in the figure. Such a figure shows the age structure of the studied population.

3. Move the first two paragraphs from the Discussion section to the Results section. This is an overview of the results and a summary results.

4. I propose to separate the section Limitations of the Study from the discussion.

Author Response

Dear Reviewer,

Thank you for your analysis of our manuscript.

Please find our answers to your questions and commentaries.

Sincerely

Stephane Olindo

Reviewer 1:

  1. Complete, if possible, size and age structure of the North Medoc population. This information is important for a complete picture of the study group.

We agree with the reviewer and also believe that knowledge of the size and age structure of the population of North Medoc would be of great interest. Unfortunately, reliable data are not available in this area. However, rates of screened population have been performed in the 3 other communities.

  1. If possible, create an age pyramid of the studied population in the figure. Such a figure shows the age structure of the studied population.

We have created an age pyramid according to sex of the population studied in the 3 communities. This figure is the number 1 and the 3 previously submitted figures are named 2 to 4.

  1. Move the first two paragraphs from the Discussion section to the Results section. This is an overview of the results and a summary results.

The second paragraph of the “Discussion” that summarized the findings has been moved to the end of the “Results” section.

  1. I propose to separate the section Limitations of the Study from the discussion.

We have separated “Limitation of the study” from the “Discussion” and then created a “Conclusion” section.

Eventually, the English language has been revised.

Reviewer 2 Report

Dear Authors

Many thanks for the opportunity to review this manuscript. Screening of atrial fibrillation outside the office is still an
unsolved problem. I have some quetions:
1. We know that 4,208 patients were screened. What was the entire
population of the examined communes?
2.
How many patients studied had an implanted pacemaker or
cardioverter-defibrillator? Could the presence of the implanted
device affect the screening result?
3.
What was the management of the patients diagnosed with new atrial
fibrillation? Did the pharmacist make an appointment with
a cardiologist?

Author Response

Dear Reviewer,

Thank you for your analysis of our manuscript.

Please find our answers to your questions and commentaries.

Sincerely

Stephane Olindo

Reviewer 2:

  1. We know that 4,208 patients were screened. What was the entire population of the examined communes?

Reliable data regarding community population were available for Pessac, Saint Medard en Jales and Arcachon. 3694 subjects were screened in these 3 communities with a pooled population 65 years of 21521 people.

In the result section the sentence has been modified as follow: In the pooled population (65 years) of the 3 communities (n=21 521), 3694 subjects have participated to the campaign giving a total screening rate of 17.2% (16.6-17.7).

2.How many patients studied had an implanted pacemaker or cardioverter-defibrillator? Could the presence of the implanted device affect the screening result?

Thank you for this important issue. We have analyzed the subgroup of patients with implantable pacemaker that represent 70 (1.7%) of the participants. A high proportion was detected positive (n=13, 18.6%) but 10 had a previously known AF and 3 had an EKG trace with artifacts. The EKG trace of the 57 negative screening participants with implantable pacemaker have been checked again and no AF pattern was retained.

For the manuscript we have given the following information about this subgroup: “A total of 70 (1.7%) participants reported wearing an implanted pacemaker. In this subgroup, 13 18.6%) were screened positive including 10 with a previously known AF and 3 positive screening related to artifacts.  No previously unknown AF was detected in subjects with implanted pacemaker.”

  1. What was the management of the patients diagnosed with new atrial fibrillation? Did the pharmacist make an appointment with a cardiologist? 

As written in the method section, individuals who were screened positive were invited to visit as quick as possible their GP with documents including information about the campaign and the ECG trace. GPs were also kept informed by post that their patient were screened positive for a probable AF.

The following sentences have been added to the method section:

“The GP was also systematically kept informed by post that his patient has participated to the Protect-AVC campaign and has been diagnosed with a probable AF. In France, the GP is the coordinator of the patient pathway and if required he refers his patient to the cardiologist.”